# Microstructure Analysis and Strength Characterization of Recycled Base and Sub-Base Materials Using Scanning Electron Microscope

**Tanvir Imtiaz** [1,*] **, Asif Ahmed** [2,†] **, MD Sahadat Hossain** [3,†] **and Mohammad Faysal** [4,†]

1   Graduate Research Assistant, Department of Civil Engineering, The University of Texas at Arlington,
    416 S Yates St., Arlington, TX 76019, USA

2   Assistant Professor, College of Engineering, State University of New York Polytechnic Institute,
    Utica, NY 13502, USA; asif.ahmed@sunypoly.edu

3   Professor, Department of Civil Engineering, The University of Texas at Arlington,
    416 S Yates St., Arlington, TX 76019, USA; hossain@uta.edu

4   Staff Geotechnical Engineer, D&S Engineering Labs, LLC, Denton, TX 76205, USA; md.faysal@mavs.uta.edu

*   Correspondence: tanvir.imtiaz@mavs.uta.edu

†   These authors contributed equally to this work.

**Abstract:** The reuse of recycled crushed concrete aggregate (RCCA) and reclaimed asphalt pavement (RAP) can provide a sustainable solution for the disposal of C&D waste materials instead of sending them to landfills. More importantly, it will conserve energy and reduce environmental concerns. Several states in the USA have been using RCCA and RAP as base materials for years, focusing on the quality of the recycled materials. The structure of Recycled Aggregate (RA) is more complex than that of Natural Aggregate (NA). RAs have old mortar adhered on them that forms a porous surface at the interfacial transition Zone (ITZ) and prevents new cement mix from bonding strongly with the aggregates. The objective of this study was to correlate microstructural properties such as microporosity, inter and intra aggregate pores with the unconfined compressive strength (UCS) of RAP and RCCA molds, mixed at different proportions. In this paper, the quantity of micropores and their effect on the strength of mixed materials is used as the basis of microstructural analysis of recycled concrete and reclaimed asphalt. Microstructural properties obtained from analyzing scanning electron microscope (SEM) images were correlated with unconfined compressive strength. Intra-aggregate and interaggregate pores were studied for different ratios of cement treated mixture of RAP and RCCA. The results show that addition of RAP considerably increases the number of pores in the mixture, which eventually causes reduction in unconfined compressive strength. In addition, significant morphological and textural changes of recycled aggregates were observed by SEM image analysis.

**Keywords:** recycled crushed concrete aggregate; reclaimed asphalt pavement; unconfined compressive strength; microstructure; scanning electron microscope; image analysis

## 1. Introduction

Approximately 2.6 million tons of recycled crushed concrete aggregate (RCCA) and 100 million tons of reclaimed asphalt pavement (RAP) are generated in the US every year [1]. Instead of landfill disposal, these materials can be used as an alternative for natural aggregates (NA) in pavement construction. Use of recycled aggregates can save energy and provide a sustainable solution for Construction and Demolition debris (C&D) waste management. Pavement rehabilitation projects generate huge amount of waste which are not accepted by the landfills. It is imperative for the pavement industry to find alternative uses for such materials. Base layer contributes most to

the structural capacity of flexible pavement systems and thus, high quality materials are essential. The quality of base course materials significantly affect the rate of load distribution [2]. Recycled materials have been reported to be a very effective solution for reducing pavement maintenance and construction costs [3]. However, compared to natural aggregates (NA), recycled aggregates (RA) are weaker [4]. As a results, when recycled aggregates are used as a substitute for natural aggregates in construction of the pavement base, the minimum requirement of strength standards designated by AASHTO and local state guidelines are not fulfilled [5]. Hence, to comply with the minimum strength requirements, different chemical and mechanical stabilization techniques are implemented [6]. Researchers have performed a significant amount of investigation to improve the quality of recycled aggregate mixes [7–9].

In the 1990s, mechanical properties of RAP have been investigated by several researchers [10]. However, a limited numbers of studies have been conducted on the microstructure and its effects on the mechanical properties of recycled concrete aggregate [11]. Recycled aggregates have microstructural features similar to that of natural aggregates, though they are considerably complex [12]. The contrast is attributed to the diversity inherent to the primary material composition and strongly dependent on processing and treatment [13]. It also depends on aggregate type and properties [14]. As RAs are broken up progressively, cement paste accumulates in the fine fraction and therefore density of fine RAs is lower than that of coarse recycled aggregates of the same origin. In addition, recycled aggregates are rougher, more irregular and more angular [13]. As recycled aggregates are more irregular in shapes due to their recycling processes, their specific surface areas tend to be higher than those of the natural ones [15,16]. Higher specific surface area is subjected to develop more bonding with cement paste as well as water absorption. Furthermore, this irregular and rough surface area is also responsible for more microvoid in between the bonds.

Microstructural characterization of recycled aggregate concrete is a powerful tool for determining the above-mentioned factors. Cement treated recycled aggregates contain more complex microstructure compared to concrete with natural aggregates. Microscopic level investigation contributes to the development of the durability and mechanical properties of the complex and heterogeneous material [17]. High microstructural complexity is found due to the heterogeneity of hydrated cementitious products. Strength and durability of the cement treated base can be affected by the porosity of the cement paste and the quality of aggregates. These materials have higher porosity than their natural counterparts and are deleterious to the formation of interfacial transition zone (ITZ) with new paste [17]. In cementitious materials water cement ratio gradient evolves around the aggregates during casting. As a result, different microstructure develops surrounding the hydrated cement paste. The area surrounding the aggregate particles is known as the interfacial transition zone (ITZ) [18]. Porous ITZs weakens the bonding between the cement paste and the aggregates. In normal strength concrete, porous interfacial transition zone microstructure can be ascribed to the higher porosity and absorption capacity of the recycled aggregate [19].

Texas Department of Transportation (TxDOT) guidelines require a minimum compressive strength of 2.068 MPa (300 psi) for the base layer of the pavement. Research has shown that RCCA mixed with up to 50% RAP can be used but must be treated with 4% to 6% cement to meet the minimum requirement [20]. Performance of cement treated recycled aggregates largely depends on the quality and origin of the recycle aggregates used [17]. Inadequate studies on microstructural analysis of cement treated recycled aggregates drives the momentum for this experimental study. Since recycled materials consist of large amounts of porous materials, investigation of their porosity is required. Porosity is related to the compressive strength of a material. Porosity can vary with distinct mixing ratios of RAP and RCCA. However, these mixtures are treated with cement that leads to the change in their chemical composition. Changes in chemical and elemental properties also can be a function of various strength parameters. Scanning Electron Microscopy (SEM) and Energy Dispersive X-ray Spectroscopy (EDS) were performed on small samples incised from UCS samples, which were prepared at optimum moisture content with different ratios of RAP and RCCA treated with 6% cement. SEM images were

subjected to micropore analysis to determine the pore percentage. Furthermore, element percentages found from the EDS were used for the characterization of the strength properties of recycled base materials treated with cement.

## 2. Materials and Methods

The test program was developed to determine the microstructural properties of cement treated recycled pavement materials such as reclaimed asphalt pavement and recycled crushed concrete aggregate. To obtain the actual picture of the microstructure of these mixtures without disturbing them, a precise methodology had to be carried out. For the experiment, five different mixing ratios of RAP and RCCA, treated with 6% cement were taken into consideration. Scanning electron microscopy and energy dispersive X-ray spectroscopy were performed on the samples. Results obtained from these tests were compared to the unconfined compressive strengths of the respective mixtures. Reclaimed asphalt pavement (RAP) and recycled crushed concrete aggregates (RCCA) were collected from the TxDOT specified stockpiles of Big City Crushed Concrete, which is located at Goodnight Lane, Dallas, Texas. Portland Type II cement was used as the binder which has a compressive strength of greater than 50 MPa (7252 psi) at 28 days. Low viscosity epoxy resin and fine sand papers were used to prepare the samples.

### 2.1. Unconfined Compressive Strength Sample Preparation

Five different combinations of RAP and RCCA were selected for unconfined compressive strength test. The cement content of typical pavements with cement treated base remains within 3% to 10% of the total dry weight of the mixture. In previous studies, 100% RCCA material met the minimum strength criteria of 2.068 MPa (300 psi) at 4% cement content. A combination of 50% RAP and 50% RCCA materials reached the unconfined compressive strength of 2.068 MPa at 5% to 6% cement content. A combination of 70% RAP and 30% RCCA materials fulfilled the minimum strength requirement of 2.068 MPa at 6% cement content [20]. Based on the results from previous studies, each of the combinations were treated with 6% cement. Table 1 summarizes the material combinations used in this experiment.

**Table 1.** Combinations for experimental program.

| Mixture Identification | | M1 | M2 | M3 | M4 | M5 |
|---|---|---|---|---|---|---|
| | | 0-100-6 | 10-90-6 | 30-70-6 | 50-50-6 | 100-0-6 |
| Combination | RAP | 0 | 10 | 30 | 50 | 100 |
| | RCCA | 100 | 90 | 70 | 50 | 0 |
| | Cement | 6 | 6 | 6 | 6 | 6 |

For a cement treated flex base material, unconfined compressive strength (UCS) is one of the important parameters in pavement design. UCS test results serve as the variations of strength and stiffness of the base material with change in mixing ratio. As labeled from M1 to M5 (Table 1), four samples were prepared for every combination containing 6% cement. All specimens were prepared according to Tex-113-E guidelines [21]. Samples were compacted in a mold with 152.4 mm (6 inch) in diameter and 203.2 mm (8 inch) in height, at optimum moisture content. According to TxDOT specifications a mechanical compactor was used to achieve required compaction. Each specimen was assembled at 4 lifts with 50 blows for each. Prior to testing, the specimens were preserved in a moisture room for 7 days at 70 degrees Fahrenheit according to soil-cement testing procedure [22] by TxDOT. Samples were subjected to a compressive load using the Universal Testing Machine (UTM) at a strain rate of 2.0 ± 0.3%. Following the test procedure of Tex-113E standard method, the ultimate load capacity of the sample was taken when it failed at maximum compressive load.

## 2.2. Scanning Election Microscopy

The scanning electron microscope technique is one of the well established methods to investigate the surface structure of materials. SEM produces images by probing the sample with a focused beam of electrons, which interacts with atoms on the surface to produce various signals that contain information about the material [23]. SEM equipment is coupled with a chemical analysis apparatus such as an energy dispersive X-ray spectroscopy (EDS). This apparatus can detect the characteristic of X-rays produced from interaction of electrons with the sample material [24]. For the experimental study, Hitachi 3000N SEM Microscope has been used to perform both SEM and EDS. Hitachi 3000N SEM Microscope was operated to examine cubed samples at low vacuum (VP-SEM) (Figure 1a). Back-scattered electron imaging (BSE) was performed to detect with a low acceleration voltage ranges from 15 kV to 25 kV. Sample cubes were mounted on a 15mm metal disk using carbon tape.

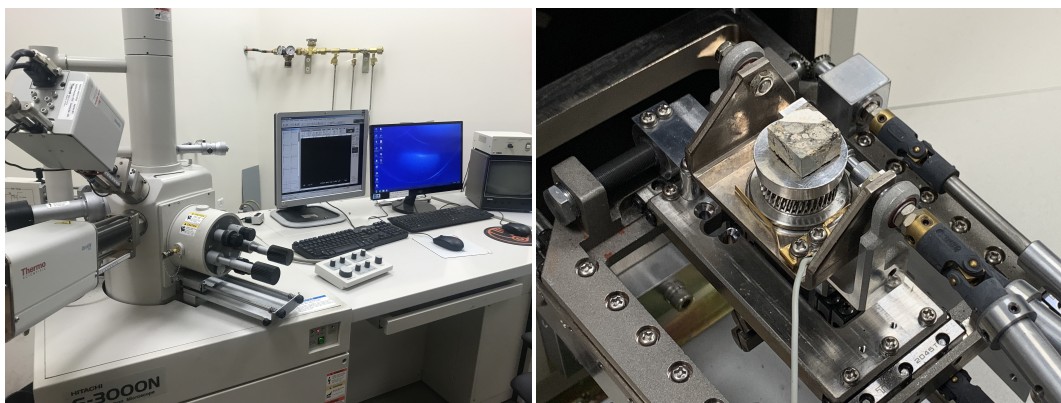

(**a**) Hitachi 3000N SEM Microscope　　　　　　(**b**) Sample inside SEM Microscope

**Figure 1.** Scanning electron microscope and sample mount for microscopic observation.

For micropore analysis, back-scattered electron images (BSE) at $100\times$ magnification were captured on a polished surface. Rather to examine microstructure of RCCA and RAP, secondary images were taken at $2500\times$ to $3000\times$ magnifications. Hitachi 3000N SEM machine is also equipped with energy dispersive X-ray spectroscopy that allows one to acquire element properties from the same SEM image. From each combination, cubes are incised randomly. Images are acquired from four different cubes for the validation of the examination.

## 2.3. Sample Preparation for Microscopic Observation

After 7 days of curing, UCS samples were tested under unconfined compression. Broken pieces of UCS samples were taken to prepare the SEM samples. Samples are immersed in isopropanol to replace the water from them [25]. Later on, vacuum suction was applied to remove the isopropanol. The samples were impregnated with a low-viscosity epoxy resin [26]. A sample was placed in a cylindrical mold; the impregnation was done under vacuum to ensure that there was no moist air inside the pores. After impregnation samples were left at room temperature for 24 h to polymerize. When the resin was completely hardened, samples were incised into cubes (Figure 2a) of about 12.7 mm (0.5 inch) on each side. Precautions were implemented to maintain the integrity of the samples during incision. Incised specimen was polished using SiC sandpaper to ensure smooth surface. Isopropanol was used as lubricant during the polishing. Finally, the samples were washed with isopropanol and kept in a desiccator to evaporate the excess chemicals [25].

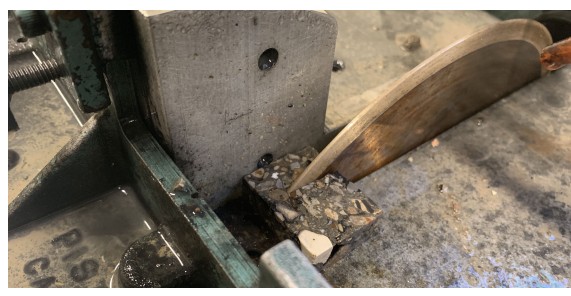 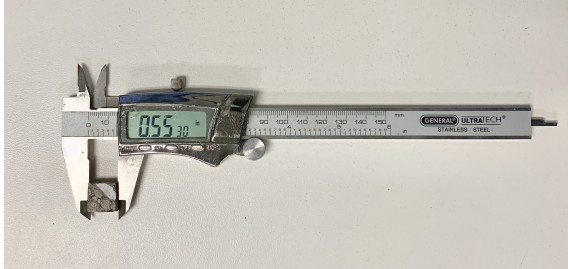

(**a**) Diamond blade saw          (**b**) Sample measurement

**Figure 2.** Scanning electron microscope sample preparation for microscopic observation.

### 2.4. Image Analysis

To measure micropores between the aggregates after curing, SEM photographs were taken at 500 μm fraction. The captured photographs were used for further analysis. Detectable pores can be identified from the gray-scale contrast. Since the images are black and white, in terms of gray-scale the darkest areas of the image are recognized as the porous areas. For quantitative analysis of SEM images, vectorization software can be utilized [27]. In this study, the images were converted to binary images using the software *imageJ*. The amount of pores can be determined by calculating the black and white area. In the image, each 1000 pixels were considered to be equal to 1mm [17]. Average value from all measured pore area for each combination was determined to be the amount of pores for that combination.

## 3. Results and Analysis

### 3.1. Qualitative Analysis

Microstructural study of different heterogeneous materials such as RCCA and RAP treated with lower amount of cement illustrates distinctive hydration products. Density, shape, and size; pore structure, stability, and strength are the main properties of the aggregates that influence concrete behavior [28]. In the microscopy study of concrete calcium hydroxide (CH), tricalcium disilicate hydrate (C-S-H), pores and residual hydrated cement are the significant components to be analyzed. Various field image analysis processes have been developed to analyze the quantity and quality of these properties [14]. C-S-H is mostly responsible for the behavior and strength of the hardened cement. RCCA and cement paste have almost similar chemical behavior [29]. RCCA consists of significant amount of fine aggregates which are basically old mortar. The presence of ettringite (spike like crystal) helps to distinguish between natural and recycled aggregates. $3CaO·SiO_2$, $2CaO·SiO_2$ and $3CaO·Al_2O_3$ and the solid solution with average composition $4CaO·Al_2O_3·Fe_2O_3$ are the fundamental components of cement [28]. During hydration, $3CaO·Al_2O_3$ reacts with gypsum $CaO·SO_3·2H_2O$ to sequentially form hydrous calcium aluminum sulfate ettringite $Ca_6Al_2(SO_4)_3(OH)_{12}·26H_2O$ and monosulfate. These are the initial reactions which occur during the first 24 h of hydration. Further reactions produce C-S-H which strengthen the material [29]. Figure 3a illustrates the presence of ettringite and C-S-H. However, RAP usually does not undergo hydration reaction due to its asphalt overlay. RAP aggregates are partially or fully covered by bituminous binders that prevent them from reacting with cement. In Figure 3b, round crooked shaped asphalt overlay incorporated with air voids, is detected.

### 3.2. Pore Analysis

Back-scatter SEM can usually detect a significant amount of visible pore areas in most cement treated compositions. Epoxy resin fills the pore spaces in prepared samples. Because of the ability of low electron back-scattering, epoxy resin filled pores appear darker than other materials in the composition [30]. So the secondary electron images are converted to 8-bit binary image in gray scale

having an intensity of 0 (black) and 255 white (white) [31]. In terms of porosity, four features of a pore system are (a) micropores in the 0.5 to 10 nm range (gel pores), (b) mesopores in the 5 to 5000 nm range (capillary pores), (c) macropores due to compaction and (d) shrinkage cracks are taken into consideration [32]. Except pores that are smaller than a pixel, the darker areas of the binary images is considered as pores. Area of pores is calculated using the *imageJ* software [14]. SEM photographs are captured from four randomly incised cube for each RAP-RCCA combinations to ensure representativity. The average percentage represents the amount of pores found in that combination. The amount of average pore percentage increased with the substitution of 0, 10, 30, 50 and 100 percent RAP material. Average pore percentages are found to be 4.85, 5.95, 8.01, 10.49 and 16.49 percent respectively. Table 2 shows the average area of pores in each combination.

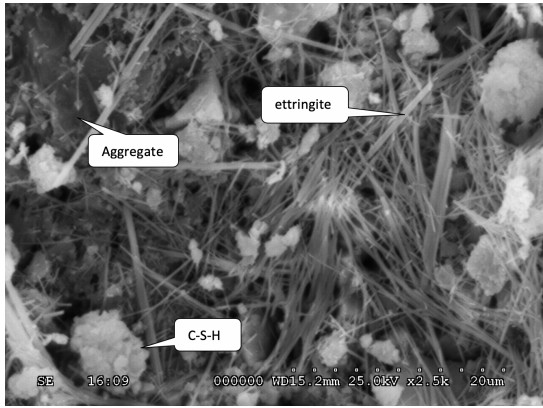 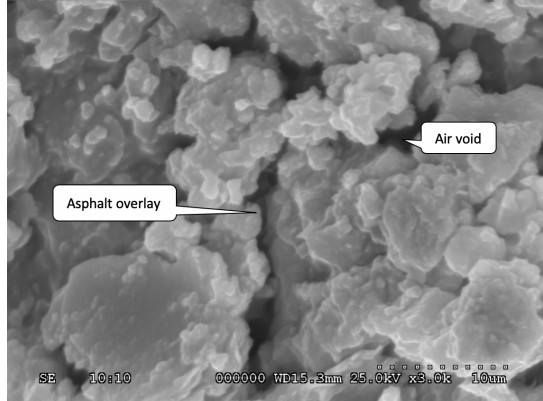

(**a**) Secondary electron image of recycled crushed concrete aggregate (RCCA)

(**b**) Secondary electron image of reclaimed asphalt pavement (RAP)

**Figure 3.** Scanning Electron Microscope image of (**a**) RCCA and (**b**) RAP.

**Table 2.** Area of pores.

| Combination ID | Area of Pores (%) | Average Area of Pores |
|---|---|---|
| M1 | 3.767 5.820 7.458 2.345 | 4.85 |
| M2 | 3.690 6.580 5.140 8.381 | 5.95 |
| M3 | 9.713 7.121 3.907 11.290 | 8.01 |
| M4 | 11.387 5.980 15.489 9.088 | 10.49 |
| M5 | 9.607 3.437 20.365 32.543 | 16.49 |

For instance, Figures 4 and 5 present the binary conversion from scanning electron microscopic image for 30% RAP, 70% RCCA and 100% RAP, 0% RCCA stabilized with 6% cement respectively. The larger pores correspond to entrapped air bubbles due to inadequate compaction. In RCCA these pores are gradually filled by hydration reaction with residual cementitious material and newly formed CH [19,28]. Asphalt overlay restricts the RAP aggregates from the reaction thus making them unable to fill the pores.

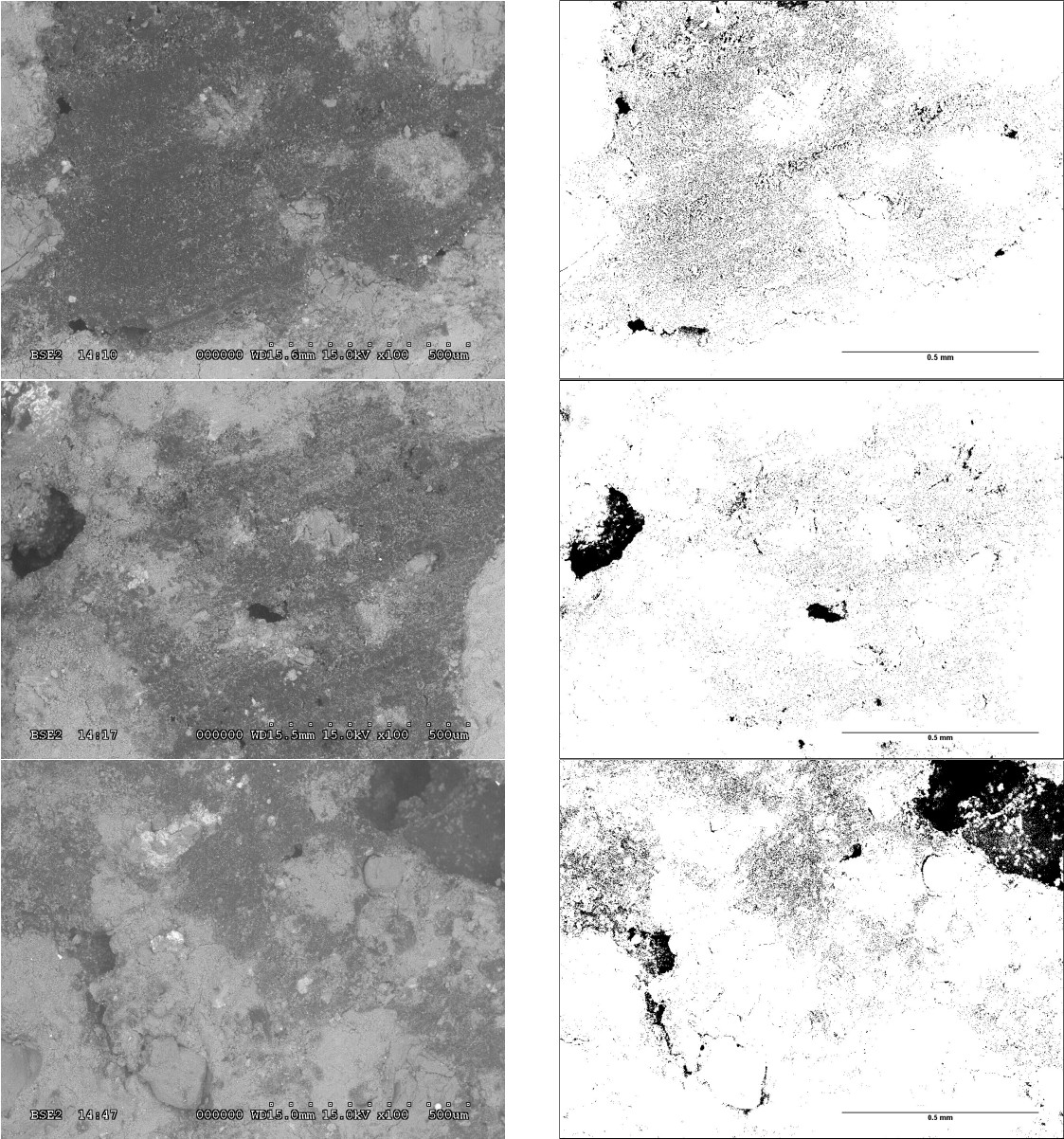

**Figure 4.** SEM (**left**) and binary image (**right**) of 30% RAP, 70% RCCA with 6% cement.

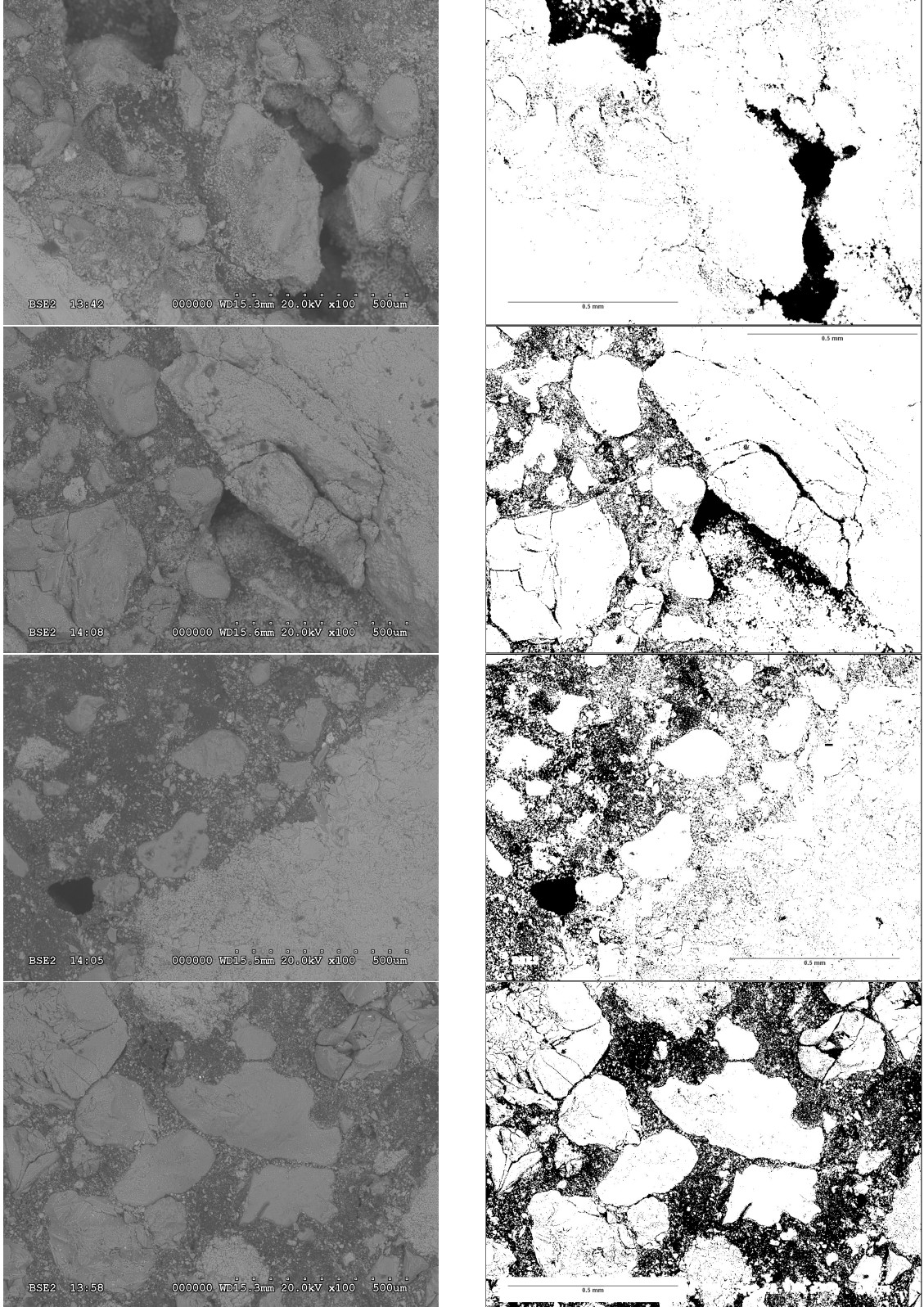

**Figure 5.** SEM (**left**) and binary image (**right**) of 100% RAP, 0% RCCA with 6% cement.

### 3.3. Porosity and Compressive Strength

Porosity has significant impact on the strength of aggregate blend for the treated base. Porous substances are generally weaker than solid objects. Compressive strength of a material is correlated to its porosity.

In Figure 6 compressive strength is represented along with the average percentage of pores for different combinations of RCCA and RAP treated with 6% cement. Compressive strength decreases with increase in percentage of pores as shown in Figure 7. Due to the increase of hydrated compounds with hydraulic properties, amount of pores reduces; thus the compressive strength increases [33]. In aggregate blends, higher percentage of RAP aggregates create more voids compared to RCCA. As hydration products tend to migrate into pores, RCCA pores get filled up. Interfacial transition zone (ITZ) is often found along the weakest part of the concrete [34]. Asphalt overlay created porous ITZ resulting in weaker bonding between them. As shown in Figure 7, the average area of pore shows a decreasing trend with increase in compressive strength, which is decreasing the intrusion of RAP material.

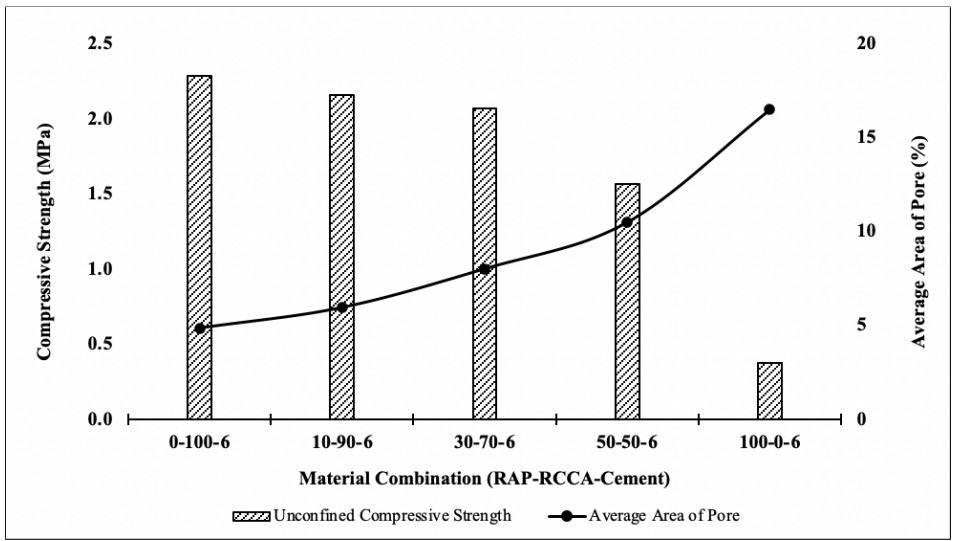

**Figure 6.** Comparison of unconfined compressive strength with porosity.

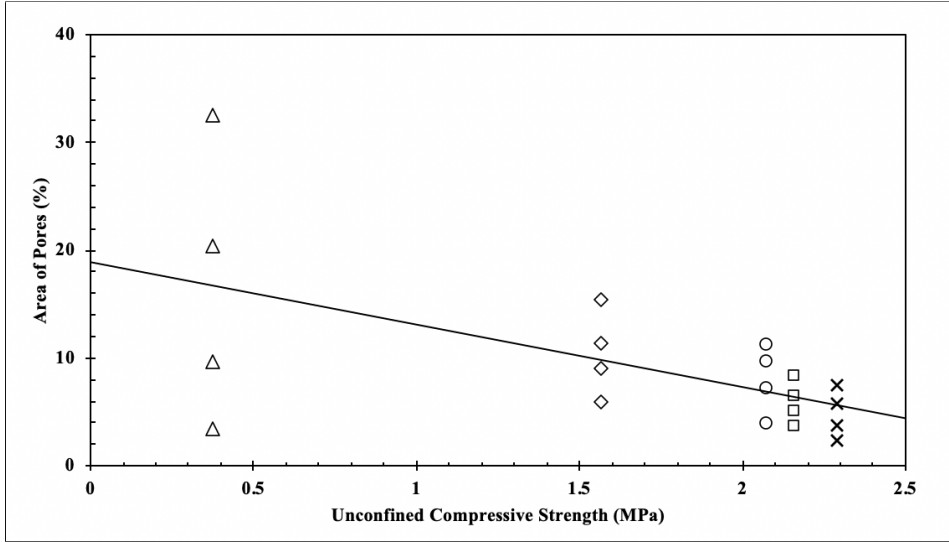

**Figure 7.** Percentages of pore area against the unconfined compressive strength.

### 3.4. Energy Dispersive Spectroscopy

Energy dispersive X-ray spectroscopy comes with the SEM and exhibits the EDS spectrum along with the average weight percentage of each element present on that image [3]. In Figure 8a, 100% RCCA treated with 6% cement demonstrates the highest count of Silicon (Si) and in contrast in Figure 8b, Calcium (Ca) shows the highest count in the spectrum. However, from the NSS software the average percentage of each element in a tabular form can be obtained. Compressive strength decreases with the intrusion of RAP material, adecrease in Calcium percentage as such 14.79, 16.59, 20.92, 26.17 and 36.68 respectively (Figure 9a) can be observed. However, the weight percentage of Si increased with increase in compressive strength as follows: 18.33, 14.93, 14.45, 12.23 and 11.16 (Figure 9b).

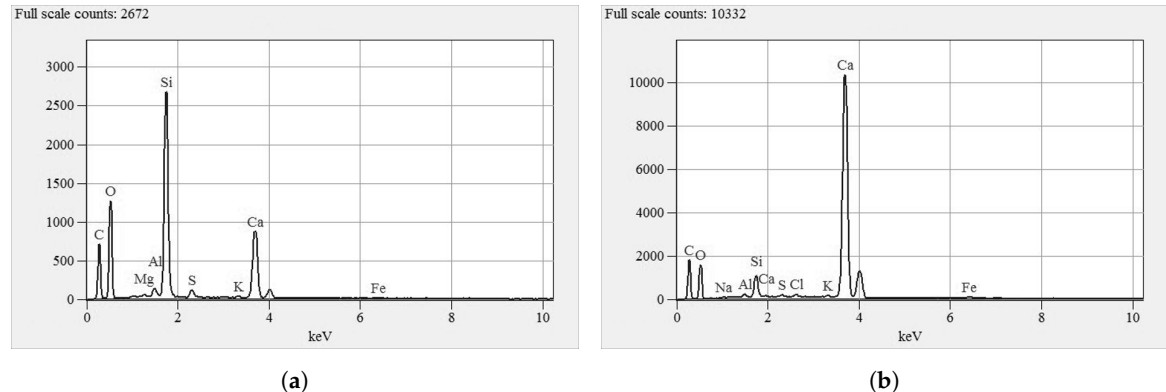

|   (a)   |   (b)   |

**Figure 8.** Energy Dispersive X-ray Spectrum of (**a**) 0% RAP + 100% RCCA + 6% cement (**b**) of 100% RAP + 0% RCCA + 6% cement.

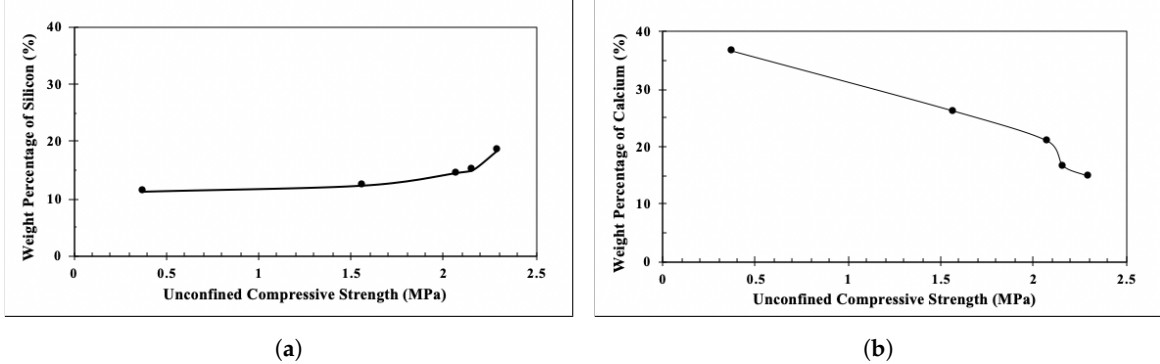

|   (a)   |   (b)   |

**Figure 9.** Weight percentage of (**a**) Silicon and (**b**) Calcium with the variation of compressive strength.

### 3.5. Ca/Si Ratio and Compressive Strength

The fundamental element of hydrated cement paste, the carbonation behavior of C-S-H mainly depends on its structure, is affected by the Ca/Si ratio [35]. Changes in C-S-H gel significantly impacts the strength characteristics of hardened cement paste. Researchers have found significant increase in compressive strength by lowering Ca/Si value of C-S-H paste whereas comparatively less compressive strength was obtained at high Ca/Si value [36]. In this experimental study, a similar trend demonstrating decreasing compressive strength with increase of Ca/Si ratio, was observed. Figure 10 represents the co-relation between compressive strength and (a) Area of pores with a $R^2$ value of 0.9701 and (b) Ca/Si ratio with a $R^2$ value of 0.9647.

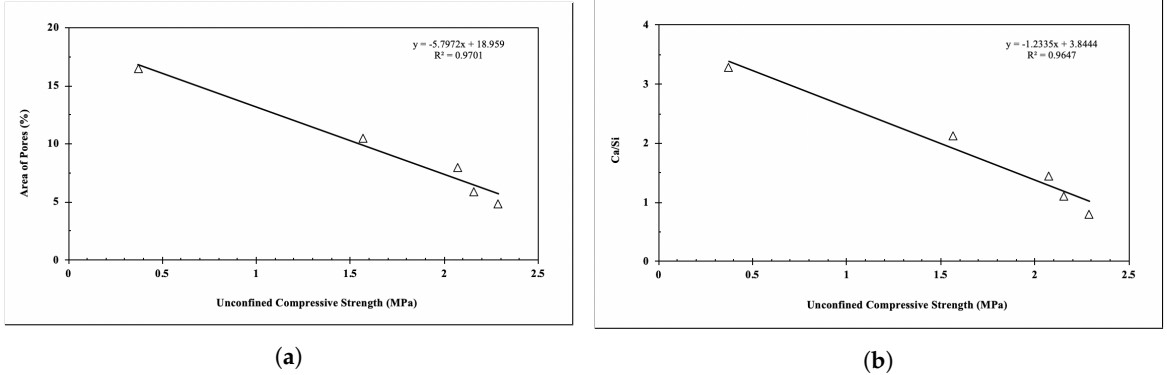

**Figure 10.** Variation of compressive strength with (**a**) area of pores and (**b**) Ca/Si ratio.

## 4. Discussion

The detection of pore of various size is limited by the resolution of SEM images. The minimum size of a pore must be the size of a dark pixel in the image. Generally, pores of sizes less than about 0.2 μm are not discoverable at the usual range of magnification. However, as indicated in previous research work, much finer pores can be seen in FE-SEM examination [37]. Nevertheless, the lower limit of conventional SEM-detectable pore sizes is usually larger than the upper limit of pore sizes reported in mercury intrusion porosimetry study of hydrated cement paste; the latter is often significantly less than 0.1 μm. Pores detected by SEM images demonstrate that commonly used mercury intrusion method underestimates the smaller pores which are present in the image [30]. The method of calculating average area of pores, used in this study, was based on limited test samples. However, the primary analysis indicates that quality of recycled aggregates can be determined. Furthermore, SEM-EDS data of different RAP-RCCA combinations at various cement content, is required to establish the corelation of pore area and Ca/Si ratio with unconfined compressive strength.

## 5. Conclusions

Based on the experimental study, microstructural analysis might be accepted as a convenient approach for the characterization of heterogeneous material such as recycled aggregates. Experimental study on the microstructural and elemental properties of different combinations of recycled aggregate blends concludes some far-reaching features that can be beneficial for further research.

- Microscopic photographs can distinguish between recycled aggregate and natural aggregates. Recycled aggregates are weaker than natural aggregates. Microstructure of recycled aggregates are heterogeneous, irregular and inconsistent. Recycled crushed concrete is progressively broken up and mostly covered by old cement mortar and fine fractions.
- Recycled asphalt aggregates are mostly covered by asphalt layer that prevents formation of new cement-aggregate bonding. Replacing greater portion of RAP in aggregate blends significantly reduces the compressive strength and stiffness.
- Porosity of hundred percent RAP blend material is around three times the porosity of hundred percent RCCA blend material. Porosity increases gradually with the increase of RAP as a replacement of RCCA.
- Compressive strength decreases linearly as the area of poresincreases. Compressive strength increases with the increase of weight percentage of silicon while the strength decreases with the increase in calcium percentage.
- Compressive strength is higher at low Ca/Si ratio but lower at high Ca/Si value. Strength increases linearly with the decrease in Ca/Si ratio.

**Author Contributions:** In this research project authors have the following individual contributions: conceptualization, T.I., A.A., M.S.H. and M.F.; investigation, methodology, software, T.I.; validation, formal analysis, T.I. and A.A.; resources, M.S.H.; writing–original draft preparation, T.I., A.A., M.S.H. and M.F.; writing–review and editing, T.I., A.A., M.S.H. and M.F.; supervision, project administration, M.S.H. All authors have read and agreed to the published version of the manuscript.

**Funding:** This research was funded by TxDOT Dallas District.

**Conflicts of Interest:** The authors declare no conflict of interest. The funders had no role in the design of the study; in the collection, analyses, or interpretation of data; in the writing of the manuscript, or in the decision to publish the results.

## Abbreviations

The following abbreviations are used in this manuscript:

| | |
|---|---|
| C&D | Construction and Demolition debris |
| EDS | Energy Dispersive X-ray Spectroscopy |
| ITZ | Interfacial Transition Zone |
| NA | Natural Aggregate |
| RA | Recycle Aggregate |
| RAP | Reclaimed Asphalt Pavement |
| RCCA | Recycled Crushed Concrete Aggregate |
| SEM | Scanning Electron Microscope |
| UCS | Unconfined Compressive Strength |

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
