# Peer review of "Microstructure Analysis and Strength Characterization of Recycled Base and Sub-Base Materials Using Scanning Electron Microscope"

_infrastructures, doi:10.3390/infrastructures5090070_

Round 1

Reviewer 1 Report

This paper presents the results of an experimental study of cement-treated base with RCCA and RAP. Overall, the study is interesting and beneficial to the audience. I suggest the authors to revise the paper one more time as there exist some typos and punctuation errors.

- There are four samples for each combination in Table 2, while in Lines 108-109, it was said that three samples were prepared.

- In Figure 6, it is not clear what the bars show (and the points/line). Also, Figure 7 is not referenced in the text.

- Lines 204-205: this statement does not seem correct as there is an inverse relationship between the area of pores and the UCS. Also, please check the statements in line 211-214.

Reviewer 2 Report

It is well known that RAP does not mix well with cement; hence the concrete shows low compressive strength. The research verifies that fact by studying the microstructure of the RAP+RCCA concrete. 

It would be interesting to see if the study considers different sources of RAP and RCCA, and that concludes the same fact. In Fig. 8(a) and 8(b), shows that the RCCA sample shows higher Si, and it is expected since the RCCA contains a higher percentage of silica-based fine aggregate. However, the amount of Ca is high for the RAP, but the study does not show why it is hight. Is the RAP aggregate is a Calcium based aggregate? Various RAP aggregate sources could show different results. 

Reviewer 3 Report

This paper presents studies characterization of recycled base and subbase materials. For this purpose, scanning electron microscope and microstructure analysis were considered. Comparisons between the results shows the number of pores in the mixture increases considerably by adding RAP, which eventually causes reduction in unconfined compressive strength.

This is an interesting paper. However, this reviewer does not recommend the publication of the manuscript in the present form because of the following reasons:

1) This reviewer think it will be useful if the authors provide some additional information on the methodology and Experimental procedures used in this study.

2) English usage and spelling should be improved.

3) The manuscript is focused on properties of asphaltic materials and needs better description of the properties of asphaltic materials in pavement design, such as linear and nonlinear viscoelastic properties.  They may use available literature such as the following reference:

  • Bazzaz, M. Experimental and Analytical Procedures to Characterize Mechanical Properties of Asphalt Concrete Materials for Airfield Pavement Applications. In Civil, Environmental and Architectural Engineering, Ph.D., University of Kansas, Lawrence, KS, 2018. p. 247.

This paper will recommend for publication if the authors consider the above suggestions to improve the quality of the manuscript. Some editing is still needed. I believe it will be done before publishing.
